# Self-Determination Theory and Workplace Outcomes: A Conceptual Review and Future Research Directions

**DOI:** 10.3390/bs14060428

**Published:** 2024-05-22

**Authors:** Kaylyn McAnally, Martin S. Hagger

**Affiliations:** 1Department of Psychological Sciences, University of California, Merced, CA 95343, USA; kmcanally@ucmerced.edu; 2Health Sciences Research Institute, University of California, Merced, CA 95343, USA; 3Faculty of Sport and Health Sciences, University of Jyväskylä, 40014 Jyväskylä, Finland; 4School of Applied Psychology, Griffith University, Mount Gravatt, QLD 4122, Australia

**Keywords:** autonomous motivation, need support, behavioral intervention, occupation, job satisfaction

## Abstract

Adaptive workplace outcomes, such as employee work engagement, job performance, and satisfaction are positively associated with physical and psychological well-being, while maladaptive workplace outcomes, including work-related disengagement, dissatisfaction, stress, boredom, fatigue, and burnout, are negatively associated with well-being. Researchers have applied self-determination theory to identify key motivational correlates of these adaptive work-related determinants and outcomes. Research applying the theory has consistently indicated that autonomous forms of motivation and basic psychological need satisfaction are related to better employee performance, satisfaction, and engagement, while controlled forms of motivation and need frustration are associated with increased employee burnout and turnover. Forms of motivation have also been shown to mediate relations between need satisfaction and adaptive workplace outcomes. Despite support for these associations, a number of limitations in research in the field have been identified, which place limits on the inferences that can be drawn. Noted limitations encompass an over-reliance on single-occasion, correlational data; few fit-for-purpose tests of theory mechanisms; and a lack of consideration of key moderating variables. In the current conceptual review, we discuss these limitations in turn, with specific reference to examples from the extant research applying the theory in workplace contexts, and provide a series of recommendations we expect will set the agenda for future studies applying the theory in the workplace. Based on our review, we make three key recommendations: we stress the need for studies adopting experimental and longitudinal designs to permit better inferences (i.e., causal and directional), highlight the need for intervention research to explicitly test mediation effects to provide evidence for theory mechanisms, and outline some candidate moderators of theory effects, including workplace context, job type, pay structure, and causality orientations. We expect these recommendations to set an agenda for future research applying self-determination theory in workplace contexts with a view to filling the current evidence gaps and improving evidential quality.

## 1. Introduction

Employees’ work engagement and job satisfaction are positively associated with their workplace productivity and reduced absenteeism [1,2] and negatively associated with stress, boredom, fatigue, and burnout [3,4,5,6]. In addition, employees reporting positive mental and physical health are more likely to be productive and less likely to disengage from work tasks or seek employment elsewhere [7,8]. In contrast, employees who work long hours or regularly feel under stress at work can experience burnout, reduced work engagement, and impaired job performance, which can impact productivity [9,10].

Employers and workplace leaders, therefore, recognize the value of developing interventions in the workplace aimed at promoting the use of key strategies and behaviors that promote employee work engagement and satisfaction and reduce disengagement and dissatisfaction. This is predicated on evidence indicating that employees who are motivated to engage in these strategies and behaviors may be more likely to stay engaged in their work, improving their overall job performance and work satisfaction [11,12,13]. Innovative work behaviors, creativity, idea-sharing, and service represent candidate workplace behaviors that have been identified as having the potential to promote work engagement and job satisfaction. For instance, employees who engage in team-based idea-sharing may feel a sense of peer relatedness and inspiration from their team members, which may influence their engagement in relevant work tasks and improve overall workplace productivity and individual-level job satisfaction [14,15]. Ultimately, workplace success is associated with these adaptive workplace behaviors, as well as job satisfaction, work engagement, and job performance, and is negatively associated with turnover and burnout [12,13,14,15].

In order to develop interventions that optimally enhance the uptake of these behaviors in the workforce, employers and leaders need to first identify the motivational determinants of these strategies and behaviors. Theories of motivation from psychological behavior change research may assist in identifying these determinants and may help guide the content of interventions aimed at promoting employee success and adaptive outcomes [16,17,18,19,20,21]. Prominent among these theories is self-determination theory, a pre-eminent theory of motivation that describes how forms of motivation relate to outcomes, functioning, and behavior [22]. In particular, the theory emphasizes the quality of motivation and the satisfaction of basic psychological needs as key mechanisms that are likely to enhance motivation and future persistence in behaviors. The theory has been adopted in many contexts, including the workplace, to identify the motivational determinants of behaviors likely to promote adaptive outcomes across contexts and the mechanisms involved [23,24].

Specifically, correlational research applying the theory in workplace contexts has identified forms of self-endorsed or autonomous motivation—a key adaptive form of motivation that is central to self-determination theory and reflects engaging in tasks for self-endorsed reasons [25]—and the satisfaction of basic psychological needs, particularly the need for autonomy, as key mechanisms that mediate the adoption of strategies and behaviors that lead to adaptive outcomes, like job engagement and satisfaction (e.g., task creativity or knowledge sharing) [26,27,28,29]. In order to encourage their employees to engage in these types of behaviors that ultimately promote workplace success and productivity, employers need to create environments that support these forms of motivation and need satisfaction in their employees. The theory predicts that it is the creation of a social environment or motivational climate by employers and workplace leaders that can foster and scaffold motivation and need satisfaction in their employees with respect to these behaviors. Specifically, it is employers’ adoption of autonomy-supportive behaviors that is proposed to create such a climate. Autonomy-supportive behaviors include providing employees with choice in task engagement, giving a rationale for assigned work-related tasks, and allowing employees to voice their opinions and have a say in how tasks are performed [30]. This proposition is supported by research indicating that interventions in which social agents display behaviors that promote greater choice, autonomous reasons, and responsibility and ownership in the workplace lead to greater employee engagement and satisfaction and do so by enhancing autonomous forms of motivation [31].

In the current conceptual review, we outline and critically discuss research that has applied self-determination theory in workplace contexts, identify key methodological and conceptual limitations of this research, and outline a research agenda to address these limitations. Our approach follows similar conceptual reviews of research applying self-determination theory in specific contexts, comprising an overview of current work followed by commentary, analysis, and recommendations regarding issues or limitations emerging from the overview [32,33,34]. We begin by critically reviewing research identifying the motivational determinants of key workplace behaviors and outcomes (e.g., job performance, work engagement, and job satisfaction) and research on interventions based on the theory to promote the adoption of key behavioral strategies aimed at promoting workplace outcomes. The review focuses on prior primary research applying self-determination theory in workplace contexts, particularly research referred to in leading conceptual, systematic, and meta-analytic reviews in this context [26,29,35] and key primary studies that have been published since [13,36]. Our focus on these contributions functions to effectively summarize trends in research applying the theory and, importantly, to identify methodological and conceptual limitations present in the extant research that affect the quality of the findings and the extent of the inferences that can be drawn from them and evaluate whether researchers have sought to address them. Such research has demonstrated support for associations among constructs from self-determination theory and workplace outcomes but has called for increased sophistication in research methods in the area [37]. There have also been observations on the over-reliance on correlational data and methods in extant research that limit the key inferences, such as directionality and causality. Accordingly, these reviews serve as the basis for identifying limitations and gaps in the current evidence and provide an evidential basis for recommendations for future research aimed at resolving these limitations and addressing these evidence gaps.

Specifically, we start by providing an overview of self-determination theory and its evidential basis and summarize prior work applying the theory in workplace contexts, particularly the established patterns of associations between theory constructs and workplace behaviors and outcomes. Next, we identify several key limitations and evidence gaps in the reviewed research applying the theory in workplace contexts, including an over-reliance on correlational data, the dearth of mechanism tests, particularly in workplace interventions based on the theory, and a lack of tests of key moderators. In doing so, we make reference to both initial and current primary research in this context, with a specific focus on research identified in key prior reviews that have signaled the extent to which these issues are prevalent in the research and studies that have since attempted to address them. Based on the review of this research, we provide recommendations to address these limitations, including the adoption of cross-lagged panel and experimental designs that afford means to infer directional and causal effects, tests of intervention mechanisms of action, and tests of moderators of theory motivational constructs on workplace outcomes to identify the contextual factors on which theory effects are dependent. We cite studies that have employed these proposed designs as exemplars that researchers might consider addressing these limitations and gaps in workplace research. These proposed designs are aimed at promoting greater precision in research, broadening knowledge on direction and causation in theory effects and how interventions based on the theory affect behavior and outcome change in the workplace, and identifying when the theory is likely to be most effective in accounting or variance in workplace outcomes. We expect these recommendations to form an agenda for future research applying the theory in this context and inform future practice, such as the development of interventions to promote better workplace outcomes among employees.

### 1.1. Self-Determination Theory

Self-determination theory makes a global distinction between *autonomous* and *controlled* forms of motivation, both of which reflect qualitative differences in motivation and eschew the sole focus on motivation intensity offered in other theories [38,39]. Autonomous motivation reflects acting or performing tasks for self-determined reasons, such as the inherent interest or enjoyment derived from the behavior or task or because it aligns with personally valued goals [38]. In contrast, controlled motivation reflects acting or performing tasks for externally referenced reasons or contingencies (e.g., rewards and praise) or out of perceived obligation or pressure from external social agents (e.g., authority figures) [38]. The type of motivation experienced by individuals engaged in behaviors or tasks is important because it is likely to determine the extent to which they will persist with the behavior or task over time and experience parallel salient affective (e.g., positive or negative emotion), psychological (e.g., psychological well-being or distress), and interpersonal (e.g., harmonic or disharmonic personal relations) outcomes arising from the performance of the behavior or task.

A further important premise of the theory is that the forms of motivation individuals experience when performing behaviors or tasks are determined by the extent to which doing so satisfies their basic psychological needs for autonomy, competence, and relatedness. The need for autonomy is defined as individuals’ need to experience their actions performed out of their own volition and choice and consistent with their genuine sense of self. The need for competence represents individuals’ need to gain a sense of agency and effectance when performing actions [40]. The need for relatedness represents individuals’ need to feel supported, accepted, and affiliated with those in the social context in which actions are performed. According to the theory, the potential for a behavior or task to satisfy individuals’ psychological needs serves to determine whether they will experience autonomous or controlled reasons for performing the behavior or task and the extent to which they will experience concomitant outcomes associated with that form of motivation [41]. Further, theorists have outlined the processes by which psychological need satisfaction relates to behavior or task performance and the experience of associated outcomes. Specifically, psychological need satisfaction may be associated with behavioral persistence and adaptive outcomes through their effects on the forms of motivation experienced when acting. Specifically, the extent to which individuals perceive a task or action will satisfy their psychological needs will determine the form of motivation they will experience when performing the task or acting and, therefore, the extent to which they are likely to persist with the task or behavior and experience concomitant adaptive outcomes, including well-being and life satisfaction [39,41].

The theoretically stipulated relations between the perceived need satisfaction of tasks and behaviors, the forms of motivation experienced when performing the tasks and behaviors, and the outcomes experienced provide a potential avenue for interventions aimed at increasing task and behavioral engagement. Specifically, theorists have suggested that creating behavioral contexts or environments that support individuals’ psychological needs is likely to promote greater behavioral persistence and adaptive outcomes and reduce maladaptive or dysfunctional outcomes. This can be achieved through social agents who are adept at enhancing the need-satisfying aspects of behaviors and downplaying the controlling contingencies. For example, studies have shown that interventions that train social agents to display sets of behaviors that support or foster the psychological needs of subordinates ultimately lead to increased behavioral persistence and adaptive outcomes in those subordinates in multiple contexts and for numerous behaviors [40,42]. When the need-satisfying aspects of a behavior or task are emphasized for individuals, and they perceive it to be satisfying their psychological needs, they are more likely to internalize the behavior or task as one that is need-satisfying and, in turn, will tend to cite autonomous reasons for performing it [43].

Self-determination theory has been widely applied to predict the forms of motivation adopted and behavioral, affective, psychological, and interpersonal outcomes in multiple behaviors, populations, and contexts, including educational, social, health, and, importantly, occupational and workplace contexts [23,24,44]. Beyond support for the associations between autonomous forms of motivation, need satisfaction, and adaptive outcomes, there has also been research testing the specific theory-derived mechanisms that underpin these associations. For example, the research has indicated that need satisfaction is linked to perceived need support from social agents, experiences of the tasks or behaviors as autonomously motivated, and the expression of adaptive outcomes [45,46]. These associations have also been proposed as reflecting the process by which support for needs relates to outcomes. Ryan et al. [47] formally specified a generalized process model that summarizes these key processes derived from self-determination theory, and similar models have been proposed and tested elsewhere [23,48]. Such models propose that perceived need support and need satisfaction with respect to a task or behavior will be related to behavioral persistence and associated outcomes through the mediation of the type of motivation experienced when performing the task of behavior [23,47,49]. For clarity, these proposed relations are illustrated in Figure 1. Numerous primary studies and meta-analytic research have provided support for these mediation models in numerous behavioral contexts, including the proposed indirect effects of need satisfaction and perceived need support on behavior and associated outcomes through the mediation of the form of motivation adopted [23,47,50]. This research has provided qualified support for a process by which need support and satisfaction relate to experienced forms of motivation and behavioral performance and the concomitant outcomes.

### 1.2. Self-Determination Theory and Workplace Outcomes

Correlational research applying self-determination theory in workplace contexts has indicated that autonomous forms of motivation in the workplace are associated with greater work engagement and job satisfaction and reduced stress and burnout [48,51,52]. Studies have also demonstrated that workers who cite autonomous reasons for performing work-related actions and tasks displayed more employee communication and mentoring behaviors [53] and had higher levels of job satisfaction [54]. On the other hand, individuals who cite controlled reasons for performing tasks and behaviors, such as monetary or verbal rewards, report lower levels of work engagement and job satisfaction [55,56]. Importantly, and consistent with a key premise of the theory, there is evidence that the satisfaction of psychological needs in the workplace is associated with autonomous forms of motivation with respect to work engagement, and need satisfaction is also associated with adaptive work outcomes, including better job performance, increased levels of job satisfaction, and workplace engagement [35,57]. Similarly, there is research testing Ryan et al.’s model in workplace contexts [35,47] contributing evidence on the mechanisms by which self-determination theory constructs relate to outcomes in the workplace and providing possible guidance to intervention or experimental research, which may further enable the inference of temporal or causal effects within the theory, particularly in workplace contexts. There is, however, a need for further model tests that move beyond correlational designs and permit better inferences of causal effects.

### 1.3. Self-Determination Theory Interventions in the Workplace

The growing convergence of research findings supporting associations between constructs from self-determination theory, including need satisfaction and autonomous forms of motivation and behavioral and other adaptive workplace outcomes, signals the potential for the development of interventions based on the theory that promote adaptive behavioral and psychological outcomes in this context. This may be through the adoption of autonomy-supportive techniques (e.g., choice provision) by social agents in the workplace (e.g., managers and supervisors) that may promote employees’ basic psychological needs and autonomous motivation and, in doing so, lead to increased adoption of productive work behaviors, including workplace creativity or task-sharing, which may promote better workplace outcomes. These proposals have been supported by research applying the theory in these contexts. Ryan et al.’s [47] generalized model has also been applied in workplace settings to identify the processes by which constructs from self-determination theory, like need support or autonomous motivation, relate to workplace behaviors and outcomes, including job satisfaction and well-being [36,58]. Specifically, employees’ perceptions of the extent to which their social agents in the workplace (e.g., employers, workplace leaders, supervisors, authority figures, and mentors) support their autonomy has been shown to be associated with adaptive workplace outcomes through increased autonomous forms of motivation and reduced controlled forms of motivation [36,59,60]. Such mediation tests provide preliminary evidence of the mechanism by which support for psychological needs by employers relates to adaptive outcomes and behavioral persistence in employees. This may signal possible constructs that could be targeted in interventions seeking to change workplace outcomes using appropriate techniques, such as social agents in workplace contexts offering employees need support. 

In addition, studies have demonstrated that employees’ perceptions of their work environment as being autonomy-supportive are related to their work behaviors (i.e., knowledge sharing and creativity) and work outcomes (i.e., job satisfaction and work engagement). Employees’ perceptions of the degree to which their work environment supports their three basic psychological needs of autonomy, competence, and relatedness may be especially important in the context of the theory [57]. For example, when employees perceive that the office environment fostered by social agents in their workplace is supportive of their autonomy, they are more likely to engage in work-related behaviors autonomously, which may, ultimately, lead to adaptive workplace outcomes [35]. Therefore, need support, particularly autonomy support, has been identified as a key strategy that employers may adopt to create environments in which their employees are more autonomously motivated toward workplace actions and tasks. For instance, employers who support their employees’ relatedness by encouraging team members to share their perspectives may create environments where employees collaborate more often, resulting in higher-quality task performance and overall improved job satisfaction [61].

Based on this cumulative evidence, researchers have developed interventions in which supervisors are trained to make their work environments more autonomy-supportive for their employees [35], consistent with autonomy-support training programs based on the theory developed in other contexts [62,63]. Interventions focused on supporting employees’ basic psychological needs through workplace leaders’ adoption of autonomy-supportive strategies and behaviors (e.g., providing employees with choice, considering employee perspectives, promoting responsibility and ownership over tasks, and limiting the use of rewards and punishments) by workplace leaders have been effective in improving adaptive workplace outcomes, such as improved job performance, job satisfaction, and work engagement [35,60]. 

## 2. Limitations of Current Research Applying Self-Determination Theory in Workplace Contexts

Despite generalized support for key predictions of self-determination theory in its application in workplace contexts [31,36,51], inconsistencies in the effect sizes of relations between the theory constructs and key outcomes in prior research and gaps in the current research have been observed. Research expressly seeking to resolve these inconsistencies and evidence gaps is needed to advance knowledge of the motivational bases of workplace outcomes. Accordingly, in the next sections, we outline these key limitations based on the extant literature, review research in the area [26,37,64], and provide recommendations for future research in the application of self-determination theory in workplace contexts. We focus on three key issues: (a) the excess use of correlational research designs that limit the extent to which directional and causal effects among theory constructs can be inferred; (b) the lack of mechanistic testing in interventions based on the theory; and (c) the potential contextual and theoretical moderators that may impact the magnitude of the effects of theory constructs on workplace outcomes.

With respect to the over-reliance on correlational data, while such designs have value in that they provide indications that theory constructs and outcomes are related, they do not permit inference of directional effects nor do they provide indications of how the constructs themselves, or the relations among them, change over time. So, the use of correlational data to support temporal or causal claims based on the theory ultimately falls short and misrepresents study findings. In the current review, we suggest the adoption of forms of longitudinal ‘panel’ research designs, including new implementations of these designs, such as random intercept panel designs, which control for covariance and temporal stability, as a potential solution to account for change in theory constructs and better enable researchers to make inferences with respect to the direction in the effects or their reciprocity [65].

In terms of the need to test the mechanisms by which interventions based on the theory impact change in behavior and outcomes in the workplace, we propose that researchers need to augment their research designs to enable tests of their intervention mechanisms of action. This entails including measures and design elements to intervention studies to demonstrate that changes in targeted workplace outcomes as a result of the intervention occur through the change or activation of the theory-based constructs involved [66,67,68]. The value of identifying such research is that it provides verification that the putative theoretical mechanism is responsible for the intervention effect. Although there is expanding literature testing such mechanisms in the context of self-determination theory-based interventions in other contexts [69], researchers have not routinely tested these mechanisms in workplace contexts [70]. We outline the importance of testing these mechanisms of action of self-determination theory-based interventions in the workplace and outline examples of study designs that will address this research gap and elucidate how interventions ‘work’ in changing behavior and outcomes.

Finally, inconsistencies in the observed size and pattern of effects of research applying self-determination theory in workplace contexts may be attributable to methodological artifacts, such as the sample size, study design, and lack of statistical power. These artifacts aside, the inconsistencies may also be attributable to conceptual- (e.g., the type of organizational structure—authoritative or hierarchical as opposed to democratic and ‘flat’—and individual differences in self-determined orientations or ‘causality orientations’), contextual- (e.g., worker level—blue-collar and white-collar workers), methodological (e.g., study design and quality and types of measure used), and sampling- (e.g., sample characteristics, like gender and age, and type of worker) moderating variables. We propose that researchers need to pay more consideration to these candidate moderators when conducting workplace research based on theory. In doing so, we outline some key research directions that may assist in resolving these inconsistencies, provide greater precision in the effects of self-determination theory constructs on adaptive workplace behaviors (e.g., creativity and knowledge sharing) and outcomes (e.g., job satisfaction and task performance), and identify the conditions (e.g., organizational structure, job type, advancement opportunities, and remote work options) and extraneous factors on which these relations are contingent with a view of elucidating the boundary conditions of theory predictions [71,72]. Further, the identification of moderators of the effect of self-determination theory-based interventions on workplace behaviors and outcomes may also assist in outlining the conditions in which interventions are more or less likely to be efficacious [73].

## 3. Moving on from a Focus on Correlational Study Designs

A substantive proportion of studies applying self-determination theory in workplace contexts adopt correlational, single-occasion research designs. Many studies seek to test hypotheses from the theory by measuring constructs, such as employee need satisfaction and autonomous motivation, and examining their unique associations with concurrent measures of behavior-related outcomes (e.g., job satisfaction, job performance, employee turnover, and creativity) [35,57,59]. Studies employing such designs are useful in identifying the size and variability of relations among constructs from self-determination theory and workplace behaviors and outcomes and add to an evidence base of research supporting theory-based relations in this context. However, such studies are also limited insofar as they prohibit the inference of directionality (i.e., the order of variables in a proposed nomological network) and causality (i.e., whether change in one variable affects change in another) in effects. For example, correlations among theory constructs and outcomes do not provide information on the direction of the effect, i.e., does the theory construct precede the outcome, or vice versa, or is the relation reciprocal? Nor do correlations inform whether a change in the construct through, for example, an intervention or manipulation affects a change in the outcome. Although other reviews have noted the reliance on correlational designs, they have typically not provided direct suggestions on alternative research methods that could be applied in research in workplace contexts to address the shortcomings of these designs. Further, while there has been a greater adoption of alternative designs, there is little evidence of a substantive shift, and the field still relies heavily on correlational designs.

In addition, single-occasion correlational designs do not account for a change in variables over time, known as covariance stability, or whether an intraindividual change in independent variables in a model over time is related to an intraindividual change in other dependent variables in the model [74]. For instance, assessing relations between constructs from self-determination theory in the workplace (e.g., relations between employees’ perceived autonomy support from their supervisors and their need satisfaction and autonomous motivation toward their work activities) only suggests that these variables share variance. The theory may specify the direction of these effects (e.g., supervisor autonomy support may precede need satisfaction and autonomous motivation), but correlational data do not provide verification. Further, the nature of the relations between these constructs may be more complex. For example, new information may come to light that alters one or both of these constructs, which could affect their *stability* over time. Such variations are not captured in studies testing these relations using single-occasion designs or even in prospective designs in which antecedent and consequent or *dependent* constructs are measured at different timepoints. Similarly, relations between the constructs may not be solely unidirectional. For example, it is possible that the effects of perceived autonomy support may not only influence psychological need satisfaction, but psychological need satisfaction may also influence need support, known as a *reciprocal* pattern of effects and indicative of a more dynamic process. Again, such effects cannot be inferred by studies adopting single-occasion correlational designs, and an application of more sophisticated longitudinal designs is needed to provide insight into these potential patterns of effect. 

### 3.1. Longitudinal Tests and Panel Designs

One means to potentially test these directional effects over time while also accounting for temporal stability and within-individual or intraindividual changes in theory constructs over time is through the adoption of forms of cross-lagged panel research designs. Research adopting these designs typically measures the variables of constructs of interest at multiple timepoints and theory-determined associations among the variables within and across timepoints are tested. The design may help resolve issues surrounding the direction of effects by assisting in elucidating theory-specified directional or reciprocal effects while simultaneously controlling for temporal stability. Although few studies have adopted such designs to test predictions of self-determination theory applied to workplace contexts, there is a precedence. For example, Galletta and colleagues [13] applied this design to investigate relations between proactive work behavior, organizational commitment, and autonomous and controlled forms of motivation at two timepoints. They found that the constructs exhibited substantive temporal stability and identified directional but not reciprocal effects of commitment on autonomous motivation and autonomous motivation on proactive work behavior [13].

Studies such as this provide preliminary evidence for the constancy and directionality of effects while controlling for stability over time, but there is a clear need for additional studies utilizing this design to develop a cumulative evidence base in support of these effects. It is important to note that panel designs have increased in complexity and sophistication in recent years to better allow the inference of directional effects while controlling for multiple stability components [65]. For example, the adoption of random-intercepts cross-lagged panel designs enables adequate control for both temporal stability and intra-individual change, that is, change within individuals over time, which is a deficit of the ‘classic’ panel design. The application of these designs is, therefore, advocated to provide fit-for-purpose tests of directional relations in applications of self-determination theory in the workplace and may also elucidate more complex effects, such as reciprocal effects [75,76].

In addition, researchers should consider employing alternative longitudinal designs to assist with other shortcomings associated with the existing research applying the theory in the workplace. Studies do not typically examine the efficacy of the theory in predicting workplace outcomes over extended time periods, that is, beyond a few months. The efficacy of self-determination theory for a long-range explanation of workplace outcomes is an open question. To address this, researchers may consider introducing longer periods between initial and subsequent measures of constructs and salient workplace outcomes and examine the moderators of these long-range effects. In addition, most longitudinal designs adopt a relatively modest number of sampling occasions, which may not sufficiently capture how theory constructs and outcomes vary between and within individuals over time. Resolution may lie in the adoption of methods in which measures of constructs and outcomes are taken with high frequency. For example, researchers have used ecological momentary assessment or experience sampling in which data are collected on multiple occasions across time (e.g., daily or even multiple times per day). Such research is more effective in capturing intrapersonal variability in theory constructs in the workplace over time and its association with predicting outcomes. While such approaches are rare, there are illustrative examples that may represent good practice for future research. To illustrate, Hogenelst and colleagues [77] conducted an ecological momentary assessment study examining the within- and between-person effects of constructs from self-determination theory and workplace outcomes. The results indicate that autonomous motivation was associated with daily job performance but not satisfaction; however, these associations did not hold at the group level [77]. More widespread adoption of designs that examine long-range prediction and sample-theory constructs and outcomes with high frequency has substantive potential to advance knowledge of the motivational antecedents of workplace outcomes, particularly with respect to their reliability and the contribution of within- and between-individual variability over time.

### 3.2. Experiments and Interventions

It is important to note that although studies adopting types of panel design permit inferences of directional effects over time while controlling for change, they do not resolve other key issues relating to theoretical inference, particularly the issue of causality, as the data are still correlational. Researchers need to adopt other study designs to yield definitive tests of causal relations among self-determination theory constructs and workplace behaviors and outcomes. Such designs include experimental and intervention designs in which manipulations or intervention techniques are introduced to the workplace population of interest (e.g., training leaders to provide autonomy support, providing workplace instructional messages that avoid controlling language) targeting change in self-determination theory constructs (e.g., autonomous motivation and perceived need support) and the effects of the intervention on subsequent dependent variables, such as workplace behaviors and salient outcomes (e.g., job engagement, job satisfaction, and burnout) assessed relative to a control or comparison group that does not receive the manipulation or intervention content.

In the context of self-determination theory, it is generally elements of the environment, such as the motivational climate engendered by leaders and salient others in authority, which are the primary means to change behavior and subsequent outcomes by affecting change in the mediating theory constructs (e.g., need satisfaction and the form of motivation experienced). In recent years, researchers have begun to systematically identify the kinds of techniques or strategies that leaders and other salient others in social contexts may adopt to change behavior, and these techniques, as well as the training programs designed to foster them in social agents, have generally been the focus of experimental or intervention research aimed at changing behavior and outcomes using the theory [63,78,79,80]. Studies adopting these types of design allow researchers to more effectively and definitively test the causal effects of self-determination theory constructs on salient behavioral and other work-related outcomes.

A growing evidence base exists testing the efficacy of interventions based on self-determination theory on behaviors and outcomes in the workplace [64]. For example, a study conducted in the context of a large Fortune 500 company demonstrated that training managers to foster an autonomy-supportive workplace environment has significant effects on the managers’ perceived autonomy support and, importantly, the need satisfaction, autonomous motivation, and work engagement of their employees [31]. Managers trained to display specific autonomy-supportive behaviors reported greater post-intervention perceived autonomy support, while employees reported greater need satisfaction and were more autonomously motivated and engaged in their work compared to employees whose supervisors were not trained to adopt autonomy-supportive behaviors [31]. These types of studies provide evidence of the effects of techniques designed to target key constructs from self-determination theory (e.g., need satisfaction and autonomous motivation) on key workplace behaviors and outcomes. Despite this growing body of experimental and intervention research, the prevalence of these study designs pales next to non-experimental research, and there is a need for further research to provide broader evidence examining theory-based workplace interventions, particularly those that focus on different populations and work contexts. 

## 4. Testing Mechanisms of Action of Self-Determination Theory Predictions in the Workplace

The aforementioned experimental and intervention research based on self-determination theory lends support to the efficacy of the intervention techniques used on workplace behaviors and outcomes as well as on constructs representing the theory-based mechanisms by which the intervention is purported to ‘work’ in changing behavior [66]. However, they do not directly test the mechanisms of action of the intervention as specified by the theory. Experimental and intervention designs have the potential for researchers to formally confirm these mechanisms and provide important data on whether the manipulation or intervention content affects changes in outcomes due to concomitant changes in the proposed theory-based constructs through mediation analysis [67,68]. The value of specifying and testing the mechanisms of action of theory-based behavior change interventions is important as it allows researchers to determine exactly how interventions work in enacting behavior change according to theory specification. Importantly, data on the proposed mechanism of action accumulates for interventions adopting the same techniques and mechanism across contexts, behaviors, and populations, and researchers will gain insight into the conditions that determine whether or not these theory-based interventions are likely to be efficacious [81].

Given that interventions based on the theory do not routinely test these mechanisms of action, researchers should consider adopting appropriate study designs to do so. Designing such studies is an elaborate process necessitating identification, operationalization, and implementation of the means or methods to change behavior through changes in the theory-based constructs purported to be responsible for the change [66,67,68,82]. Studies need to show that interventions designed to target change in behavior based on a theory should not only lead to changes in the behavioral outcomes of interest but also exact concomitant changes in the measures of the psychological constructs implicated in the mechanism [67]. This necessitates identifying and explicitly specifying the mechanism, that is, the constructs that, according to the theory, need to change in order to exact a change in behavior and the behavior change methods or techniques likely to lead to change in those constructs [83,84]. It also requires the selection of appropriate methods to deliver the methods to the population of interest, develop measures of the behavioral outcome [85] and the constructs implicated in the specified mechanism [83,84], and adopt an appropriate design (e.g., experimental or randomized controlled intervention design) with appropriate measures and analytic strategies to test the mechanism [66,67,68].

While there is an emerging trend to test intervention mechanisms of action in research adopting other theories of behavior change, there is still an acknowledgment that such tests are not yet the norm [70]. Testing these mechanisms of action in studies applying self-determination theory is also rare, particularly in the workplace context. However, the number of tests of intervention mechanisms of action based on the theory is increasing, particularly in the health domain. For example, Sheeran et al. [86] conducted a meta-analytic synthesis of research adopting randomized controlled designs testing the efficacy of theory-based techniques on behavioral outcomes in the health domain with concomitant changes in constructs representing the mechanisms of action to establish the appropriate mediating effects. The research demonstrated that interventions adopting intervention techniques targeting changes in autonomous motivation and perceived competence led to adaptive changes in health behavior and, critically, that the effects of these interventions were mediated by changes in the requisite constructs stipulated in the theory. This research represents an important step forward in evaluating the mechanisms of action and provides robust evidence to support two key mechanisms. That said, the research still has its limitations, as there were insufficient data to provide comprehensive evidence on the potential moderators of these mechanisms. However, as the research in the field accumulates, such moderating effects can be elucidated through similar syntheses that evaluate the mechanisms at different levels of the key moderator variables.

Research testing the mechanisms of action of the kind conducted in the health domain is needed in applied self-determination theory research in workplace contexts. Importantly, there is a need for studies to examine the effects of self-determination theory techniques (e.g., training workplace leaders to display need supportive behaviors) on subsequent changes in the adoption and maintenance of key behaviors in employees (e.g., creativity, collaboration, and knowledge sharing) and relevant adaptive outcomes (e.g., work engagement, job satisfaction, and job performance) and, critically, concomitant changes in the theory constructs implicated in the mechanisms of action (e.g., perceived need support and autonomous motivation). As the research evidence accumulates, researchers will be able to build a database of efficacious self-determination theory-based intervention in this context as well as begin to elucidate the potential conditions in which these intervention effects and mechanisms are likely to operate, such as population characteristics and types of behavior and outcome.

## 5. Candidate Moderators of Self-Determination Theory Predictions

Although research applying self-determination theory in workplace contexts has offered insight into the motivational correlates of workplace behaviors and outcomes [31,36,51] and has provided possible indicators of intervention techniques and methods that workplace leaders can adopt to change behavior and outcomes in these contexts, the size or ‘strength’ of these effects varies considerably across studies. Such variability may be attributable to the different behaviors, contexts, and populations targeted in these studies or the presence or absence of other conceptual or theoretical constructs that moderate study effects. Although some studies have examined the effect of various moderators (e.g., flow and autonomy-supportive environments) on the relations between constructs from self-determination theory and workplace outcomes [35], the formal testing of how theory effects vary according to these candidate moderators may provide further elucidation of the extent to which theory effects are generalizable or assist in identifying the auxiliary assumptions or boundary conditions on which they depend [87,88]. Researchers have also suggested the need to consider moderators when testing and evaluating the mechanisms of action of theory-based interventions to identify the extent to which mechanisms apply across contexts and other conditions that may affect when theoretical mechanisms operate [67].

In order to elucidate these potential moderating effects, researchers should seek to adopt appropriate study designs in which theory effects are tested at different levels of the identified moderator variables. For example, researchers may adopt experimental or quasi-experimental designs in which specific study or sample moderators are manipulated or varied and their effects on relations between theory constructs and behavior and outcomes investigated. Researchers could alternatively adopt designs that provide comparator groups defined by levels of key moderators. An additional means to test moderator effects is through research synthesis, such as a meta-analysis, where the effect sizes among study constructs or model constructs can be compared between groups of studies classified according to levels of a moderator. 

Numerous variables may serve to moderate the effects of self-determination theory constructs on outcomes in the workplace. Prime candidates may be environmental factors, such as workplace context, employee hierarchy, or workplace opportunities for advancement. For instance, opportunities for advancement or promotion may moderate the effects of need support on workplace outcomes. The impact of autonomy support on work engagement may be diminished in employees who see no room for professional advancement or promotion in their workplace compared to individuals with ample opportunities for growth. Individual-level constructs may also be key moderators, including personal interests, passion for work or the job, and goal orientations. For example, an employee’s perception that their job is aligned with their personally endorsed goals may act as a moderator of these relationships. The effect of perceived need support on job satisfaction may be increased in employees whose jobs are aligned with their personal interests compared to those for whom there is a misalignment. Moderator tests may, therefore, assist further in elucidating the processes by which constructs from self-determination theory relate to workplace outcomes, and researchers should identify and test moderators that are likely relevant to the workplace and employment context of interest.

Next, we discuss some candidate environmental (i.e., work context, job type, and pay structure) and conceptual (i.e., causality orientation) variables expected to moderate self-determination theory effects in workplace contexts. We provide the conceptual basis of the moderator effects and their theoretical importance and suggestions on how researchers might go about testing them. We also provide some examples of research testing these moderators that serve as examples that future researchers may follow when testing these and other moderators of the theory when applied in this context. It is important to note that the variables identified here represent only a selection of potential moderators. Theorists and researchers should be encouraged to identify and test further candidate moderators using similar research designs to the examples reviewed here in order to build a database of conditions likely to affect theory effects in their workplace contexts.

### 5.1. Work Context: Flexible Work Practices

The work environment is likely to be a candidate moderator of self-determination theory effects. A prominent example is the advent of flexible work practices adopted by some organizations as an alternative to traditional office work. Recent trends in workforce practices have meant that workers have been provided with greater flexibility over where and when they work, such as flexible work hours, the opportunity to work at home, or split time between an office or other workplace context and home. Beyond the general shift toward these flexible work patterns to reduce commuting time and promote better quality of life and work-life balance, events, such as the COVID-19 pandemic led to these practices being adopted more broadly due to mandated quarantine and stay-at-home orders [89]. 

These kinds of workplace practices may be important in the context of the application of self-determination theory for a number of reasons. For example, employees who work from home may experience greater autonomy and need satisfaction than those who work in traditional office settings. While this flexibility may afford a greater degree of taking ownership and responsibility to facilitate autonomous motivation, work flexibility may also interact or work in tandem with other practices, including workplace training. For instance, workers offered flexibility but without training to provide them with competence in managing their work according to their own schedule may experience need frustration instead of satisfaction. Conversely, employees who work in an office setting may experience higher levels of relatedness than those who work remotely, as employees working remotely are less likely to feel a sense of belonging or being a part of a team, which has been shown to decrease job performance [90]. Therefore, the workplace environment may impact the extent to which individuals perceive their psychological needs have been satisfied, which, consistent with theory predictions, would be expected to impact their motivation to engage in productive workplace behaviors and ultimately achieve successful work outcomes. Thus, the workplace context may be an important moderator of theory effects in the workplace, and designing studies to test these predictions may provide important information for employers considering the effects of flexible work practices or work-from-home options on employee motivation and outcomes. 

### 5.2. Job Type: Blue-Collar vs. White-Collar Jobs

Another potential moderating variable is job type. Specifically, the strength of the relationships between constructs from self-determination theory, such as the association between psychological need satisfaction and workplace behaviors and outcomes, may vary according to whether the target population is employed in blue-collar or white-collar jobs. *Blue-collar* jobs typically refer to jobs in trade industries that require physical skill, expertise, or labor, while *white-collar* jobs typically refer to jobs in professional or semi-professional corporate settings [91]. Due to the pervasive differences in job trajectory, managerial structures, levels of responsibility demanded, and corporate oversights inherent in these types of jobs, this contextual difference may affect the effects of self-determination theory constructs on outcomes in the workplace.

More specifically, an individual’s job type may interact with their work environment to support or thwart basic psychological need satisfaction, which may impact work engagement and outcomes. In some cases, job type may not alter theory effects. For example, workers in both blue- and white-collar jobs could have their competence enhanced when given the opportunity to master new skills in the workplace. In contrast, differences in theory effects across job types may be observed. Take, for example, the extent to which these different job types likely offer opportunities for workers to take ownership of work tasks. Blue-collar workers may have less flexibility, say, and control over their tasks and may, therefore, not have the opportunities to have a say to provide input into the tasks they perform, and their tasks may be largely determined by their manager or another agent in the workplace. Such experiences may undermine employees’ psychological needs, which may lead to reduced adaptive workplace outcomes (e.g., job satisfaction and performance) and increased maladaptive outcomes (e.g., stress and burnout) [92]. Alternatively, employees in positions that offer greater responsibility may have greater control, flexibility, and say in their job, which provides them with more opportunities for need satisfaction that may lead to better work-related outcomes. Job type may, therefore, moderate the effects of need satisfaction on workplace outcomes. This may signal to employers that interventions aimed at supporting basic psychological needs in the workplace, such as providing opportunities to take ownership and have a say in tasks and workplace practices, may be especially pertinent among workers employed in blue-collar jobs.

### 5.3. Pay Structure: Performance-Based Pay vs. Regular Salary

Pay structure is another notable variable that may moderate relations between self-determination theory constructs and workplace behaviors and outcomes. Specifically, employees who work for a commission or on other performance-related pay structures may be motivated differently than employees who receive an hourly wage or those who receive a salary. Understanding how motivation may differ across workers under these types of pay structures may affect the extent of individuals’ need satisfaction and their subsequent motivation, behavior, and work-related outcomes. 

Specifically, the way that pay structure interacts with the work environment or leadership style may impact how employees perceive need support in the workplace. Although pay inherently incentivizes work performance, autonomous motivation can be enhanced or undermined in either pay structure. For example, the type of feedback offered to accompany a paycheck may act as the determining factor in how pay either supports or undermines autonomous motivation. If pay is presented as contingent on performance alone, it is likely to undermine autonomous motivation [93,94]. However, if pay is presented as merely indicative of workplace success in conjunction with progress toward more autonomous or self-endorsed goals (e.g., a supervisor indicating that the commission is because the worker is improving their skills in marketing the product or is achieving a personally set target) then receiving pay is likely to enhance autonomous motivation. In addition, some pay structures may more readily lend themselves to providing informational feedback (e.g., bonuses or commission-based positions), which may give employers a greater opportunity to intervene with a message about improvement than a regular wage.

Acknowledging how different environments present varied opportunities for autonomy support via the differentiation in pay structures may provide researchers with information about key targets for interventions aimed at bolstering autonomous motivation in the workplace. Furthermore, pay structure may impact how variables from self-determination theory work in the context of the proposed model. For example, for individuals whose environment or pay structure influences their perception of their psychological needs being met, their motivation and consequent engagement in workplace behaviors may be impacted. However, pay structure may not be the only incentive-related moderator of these relationships. For instance, employee prestige, opportunities for promotion and advancement, and consistency with one’s interests may all represent work context characteristics that have the potential to act as moderators. Researchers should consider how worker or employer characteristics, including pay structure or other employee motivators, moderate the effects of constructs (e.g., perceived autonomy support and feedback from employers) on workplace behaviors and outcomes.

### 5.4. Causality Orientations

Intrapersonal variables (i.e., personality traits and individual differences) may also serve to moderate the effects of self-determination theory constructs on behavior and outcomes in the workplace. Causality orientations, the degree to which individuals perceive events around them to emanate from themselves or external events [16,95], may be a prime candidate moderator. Individuals who score highly on scales measuring autonomy causality orientation are more likely to interpret the actions or behaviors of others in their environment or external contingencies as need supportive and autonomously motivating, while those who score highly on control causality orientation are more likely to interpret the actions of others or environmental contingencies or unsupportive of their needs and extrinsically motivating. These orientations, therefore, have been identified as key moderators of certain predictions of self-determination theory, such as the effect of environmental contingencies on experienced forms of motivation. For example, research has indicated that autonomy-oriented individuals are less likely to experience the undermining effect of external contingencies, like rewards, on their intrinsic motivation [96].

Applying causality orientations as moderators in self-determination theory research in the workplace, employees who endorse an autonomy causality orientation may have a greater propensity to interpret workplace contingencies and situations as supportive of their needs, even those that do not ostensibly offer a high degree of autonomy support. So, autonomy-oriented workers may have greater resilience when it comes to the undermining effects of controlling contingencies on their autonomous motivation. They may, therefore, be more likely to experience their work as need supportive and autonomous, even in ostensibly controlling contexts. In contrast, those endorsing controlled orientations may be more vulnerable to having their needs undermined, particularly in contexts where support is not provided or ambiguous. Acknowledging this moderating effect of causality orientations identifies a key avenue for research and intervention in the workplace. Individuals who likely endorse a controlled orientation may be those who benefit most from autonomy-supportive interventions, and researchers would do well to screen employees for their causality orientations and target interventions accordingly.

## 6. Conclusions

Self-determination theory is a leading theory of motivation that has been applied in multiple contexts and demonstrated efficacy in accounting for behavioral persistence and adaptive outcomes [23,47,49], including workplace behaviors and outcomes [35,90]. Specifically, the research has indicated that interventions aimed at enhancing workplace leaders’ autonomy support can subsequently impact employees’ basic psychological need satisfaction, motivational orientation, and, ultimately, their persistence in adaptive workplace behaviors, resulting in more favorable workplace outcomes [90]. However, despite these findings, evidence gaps remain in the literature. In this article, we set out to review the extant research applying the theory in this context with the goal of identifying these gaps and, in doing so, making specific recommendations for researchers on how to address them, which we expect to serve as an agenda for future research. Specifically, our review spanned both initial and recent applications of self-determination theory in the workplace [35,36,37] but focused on the research reviewed and identified trends and shortcomings in prior conceptual, systematic, and meta-analytic reviews, and the subsequent primary research studies conducted since.

Foremost among the gaps identified in the research applying self-determination theory in the workplace is the lack of evidence enabling the drawing of definitive directional and causal inferences in tests of relations among theory constructs and outcomes. Restrictions on these inferences are primarily due to an over-reliance of studies in the field on single-occasion correlational designs. To address this concern, researchers should consider employing alternative designs, including variants of cross-lagged panel designs and experimental or randomized controlled intervention designs. These designs can help overcome these limitations by allowing better inference of directional and causal effects among theory constructs and workplace outcomes, both of which may increase the value of the evidence in support of theory predictions and for informing intervention development. For example, better causal inferences mean that interventionists can be more confident that adopting a theory-based technique purposed to change a particular construct can lead to changes in workplace behaviors or outcomes. We reviewed research adopting these designs and how they address concerns over inferences but also noted a marked dearth of studies in the workplace context and advocate accordingly for more research adopting these designs in this context.

An additional limitation of the extant research applying self-determination theory in the workplace is the lack of formal tests of the mechanisms by which interventions based on the theory operate in changing behavioral and subsequent outcomes. Researchers need to explicitly specify and test these mechanisms by adopting designs that allow them to test whether the expected intervention effects occur through change in the putative theory-based mediators. Such tests will provide essential information on the extent to which the underlying theory accounts for intervention effects as specified. This has important ramifications in assessing whether interventions based on the theory are not only efficacious, but sufficiently translated into effects on the desired constructs representing the theory-stipulated process. This approach can provide insight into how and why interventions work and guide the development of more effective and efficient workplace interventions based on the theory. 

It is also important to consider key contextual and intrapersonal variables that may act as moderating variables of relations between constructs from self-determination theory and behavior and outcomes in the workplace. Specifically, workplace environment (e.g., job type: blue- vs. white-collar workers and pay structure) and worker-intrapersonal characteristics (e.g., causality orientations) may impact the degree to which constructs, like need support and autonomous motivation, impact workplace engagement and subsequent outcomes. Researchers should measure these and other moderators likely to influence the relationship between self-determination theory variables and workplace outcomes when conducting research in this area. Understanding specific job- and environment-level factors that may impact intervention effects can help tailor interventions to specific workplace contexts and maximize their effectiveness.

Overall, the current evidence indicates the appropriateness of self-determination theory to identify the motivational antecedents of adaptive workplace behaviors, like knowledge sharing and creativity, and successful workplace outcomes, including job satisfaction, work engagement, and job performance. The evidence on the viability and application of the theory, however, may be limited by an over-reliance on data derived from single-occasion correlational designs, a lack of testing of the intervention mechanisms of action, and the often unacknowledged impact of moderating variables. Researchers should consider developing studies and adopting appropriate research designs that address these concerns in order to provide new data that seek to resolve these concerns.

## Figures and Tables

**Figure 1 behavsci-14-00428-f001:**
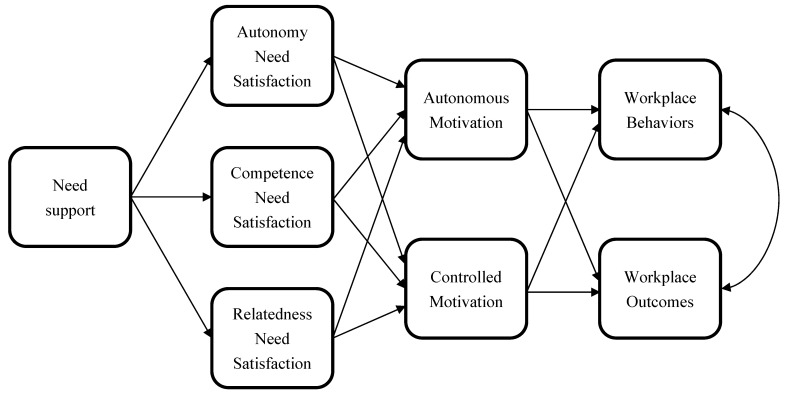
Model proposing associations among constructs from self-determination theory and workplace outcomes; adapted from Ryan et al. (2008) [47] and Slemp et al. (2018) [35].

## Data Availability

No original data were used in this manuscript.

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
