# Peer review of "Self-Determination Theory and Workplace Outcomes: A Conceptual Review and Future Research Directions"

_behavsci, 2024, doi:10.3390/bs14060428_

Round 1
Reviewer 1 Report
Comments and Suggestions for Authors
· It seems to me that the beginning of the introduction is very similar to the one in the abstract. I would try to edit it.
· I would try to revise this sentence by making it clearer: “Workers with good health more likely to be productive and less likely to disengage or seek to work elsewhere [7, 8].”
· Please add a reference for the following sentence "This proposition is supported by research indicating that interventions in which social agents display behaviors that promote greater choice, autonomous reasons, and responsibility and ownership in the workplace lead to greater employee engagement and satisfaction, and do so by enhancing autonomous forms of motivation."
· I would try to create a separate section in which to dwell on the effectiveness of the interventions, avoiding reporting them in the section called "Self-Determination Theory and Workplace Outcomes."
· Please add references for the following sentences: “A substantive proportion of studies applying self-determination theory in workplace contexts adopt correlational, single-occasion research designs. Many studies seek to test hypotheses from the theory by measuring constructs such as employee need satisfaction and autonomous motivation and examining their unique associations with concurrent measures of behavior-related outcomes (e.g., job satisfaction, job performance, employee turnover, creativity).”
· In the first part of the section "Experiments and Interventions," the authors report correct but very general statements. In my opinion, it would be better to refer them to the specific theory under consideration, in terms of past literature that has already used this approach, but also in terms of possible suggestions in using this research technique.
· A similar argument can be made for the contents given on page 10 from lines 453 to 468.
· Although no work in which the role of possible moderators is tested is actually reported, there is an entire section devoted to them. However, there is no reference to the role of possible mediators. If there is a specific explanation, it might be useful to report it.
· An additional aspect on which there was little focus in the paper concerns the type of outcome analyzed (e.g., performance, well-being).
· I think the following paragraph can be extended or revised: "Pay Structure: Performance-Based Pay vs. Regular Salary." A more general discussion can be made here about the type of motivational driver (not just limiting to salary). One could dwell on different aspects such as: prestige, opportunity to advance one's career, consistency with one's interests.
· I am not totally convinced that the way the paper is structured is as effective as possible. In particular, I believe that presenting self-determination theory after several results related to it have already been presented in the first part of the introduction may not be the best strategy. At the same time, I believe that the paragraph called "Self-Determination Theory and Workplace Outcomes" can be revised, as it contains results whose methodological matrix (correlational, panel, experimental) is not specified. And considering the subsequent focus on such aspects, it might be worth thinking of a way to integrate this paragraph with the following ones.
Comments on the Quality of English LanguageThe overall quality of the English is good, with only a few minor issues that need checking.
Author Response
Please see our responses in the attached document (which also includes responses to the comments of all the other Reviewers' comments for transparency).

Reviewer 2 Report
Comments and Suggestions for Authors
Thank you for the opportunity to read your manuscript entitled “Self-Determination Theory and Workplace Outcomes: A Conceptual Review and Future Research Directions”.
I accepted the review request with curiosity and interest. In general, I do agree with the underlying point of this body of research historically building on rather narrow research design, represented by a whole lot of cross-sectional studies. Yet, I have some concerns when it comes to the depth and detail that this review is able to address this potential issue to move the field forward. Firstly, I question what type of review the authors set put to do. The review part of the manuscript appears rather shallow, especially seen in light of the many reviews/overviews that have been done of SDT in the work domain in the past few years (see for instance, Coxen et al., 21; Deci et al., 2017; Forest et al., 2023; Manganelli et al., 2018; Olafsen & Deci, 2020; Slemp et al., 2021). The review presented here does not seem to build on such past overviews or reviews, it is really more surface-like and does no acknowledge the advancements that indeed have been made in the field since the Deci et al. review in 2017. I believe the field has indeed been advanced with more sophisticated methods in the past 5 or so years and acknowledging this in light of the critique of the literature would be only fair and beneficial. Second, since the review part in and by itself appear quite superficial what could really be a contribution of this piece is how it addresses the methodological gaps. However, as I see it, also this discussion is rather superficial and does not offer much detail in terms of applying methodological insights to this field of research in particular. Rather, it is more stating that on a general level we should shy away from cross-sectional research and rather do longitudinal studies and interventions. These are not really new insights as it appears in cross-sectional studies’ limitation sections all the time. Moreover, while I do not necessarily disagree with the sentiment, there are many layers to this. First of all, it is not always the case that cross-sectional studies are outperformed by longitudinal studies – it would rally depend on the research question at hand (see for instance Spector, 2019). Furthermore, if such a method discussion should be a contribution to this field, I would have expected a more precise discussion of research questions to be targeted, more specific design suggestions (longitudinal studies can be many different things) and also a discussion of timeframes, settings etc. Again, just doing a longitudinal study with no consideration of time lags etc. does not necessarily offer much above a cross-sectional design. Adding to this, past longitudinal studies show that may of the SDT constructs are rather stable, so more specific guidance on how to go about time lags etc. when doing longitudinal design would be appropriate for this to add a contribution to the field. The same goes for the discussion about interventions – acknowledging that it indeed have been done some in this field and rather than just stating a gap in the literature, offering a more detailed and informed discussion of how to go about this – which questions to target, in light of previous interventions, what to consider to make them effective etc. Just merely stating that we need interventions does not offer researchers in this field a lot to build on.
In sum, with a slightly different focus, stating the type of review and following the “recipe” for this, make sure the review really represent this field of research by acknowledging the indeed many longitudinal studies and some intervention studies the later years and based on this, more importantly, be very specific and illustrative of the unanswered research questions that future research can look at and the appropriate research designs considering different longitudinal design, time lags etc. would make this a welcome contribution to the field. To achieve this, this, I think that much more work is needed. These are my general comments, a few more specific ones follow.
In the abstract and early introduction, it appears that the authors are confusing terminology quit a bit. For instance, in the first sentence of the abstract the authors state that work engagement … is associated with …. engagement. Something of the same appears on page 2 in the beginning of the second paragraph starting on line 50. It is especially the term engagement that is used interchangeably. Also note that according to a few of the meta-analyses on SDT in the workplace, work engagement is seen as a wellbeing outcome, which does not align with how the authors use the term in the first paragraph of the introduction. The authors also seem to confuse need thwarting with need frustration (the former being the opposite of need support as a contextual antecedent according to the SDT model of work motivation in Deci et al., 2017, while need frustration being on the same line as need satisfaction in the same model). Also, what are outcomes vs. functioning and behavior? Looking at past overview models of SDT in the work domain. Outcomes are usually the general categories including work behaviors, work attitudes and well-being – all pointing to employee functioning? In sum, being clear and consistent within the paper as well as when talking to the past literature would be helpful in such a review.
Page 2 line 57-60, I cannot make sense of.
I wonder whether the authors can provide a reference for the claim that it is the need for autonomy that is particularly important for autonomous motivation. Take a look at the meta-analysis by Van den Broeck et al., 2016 for instance.
Elated to my general introductory comment on the type of review – on the bottom of page 2, it would be helpful f the authors could explain a bit more about the type of review they set out to do and the procedures they have undertaken in this. Also given that this is a very large field of literature, it seems important to specify what studies they have included and how these are a good representation of this field. A bit related, to me the summary of the field on page 4 seems like a mix of a review of SDT as it applies to the work context and SDT in general. If this is a review of the former, the authors should probably stick with a review of studies conducted in this domain in the review. Also, related to the next comment, I am missing an acknowledgement of later reviews in this field. Acknowledging these and also stating what the current ones add over an above these would be appropriate.
In the first paragraph on page 3, the authors outline what they are aiming for in their study. What I am missing in this part is a reasoning being the relevance and importance of these endeavors. Some of them are maybe a bit self-evident, while other are not. Either way, I think it would benefit the paper to be more specific of what the specific contributions are.
It is quite a few instances of repetition throughout the manuscript. See for instance the first part under the heading on page 5 – I think I have read this a few times already. See also paragraph one and two on page 6.
It is much to say about flexible work practices, and it is not clear cut how this affects employees. To my knowledge, it is not a whole lot of studies of remote work/telework/flexible work that looks at the effects on the associations in the SDT model, and it is not clear which role flexible work practices might have in this model (as a moderator, independent variable etc.). Thus, I would probably argue for making the discussion of these work practices more nuanced.
When the moderators are discussed on page 12, it seems like some of the examples contain more direct links that moderating links.
I hope my comments are useful for the authors in continuing develop their idea.
Comments on the Quality of English LanguageNo comments.
Author Response

(The authors gave the same response as above.)

Reviewer 3 Report
Comments and Suggestions for Authors
The article's subject is interesting, and the text is written in a pleasant style, easy to read and understand.
What I did not understand is the way the authors went about reviewing the literature. The article lacks a research methodology that should indicate how the authors proceeded to ensure that they reviewed all relevant articles on the topic studied. For this reason, the „Limitations of Current Research” section is not justified by the methodology. The authors should demonstrate that they reviewed all relevant articles and that they actually found these limitations.
The 103 bibliographic sources must be systematically presented in the paper, using tables that provide an easy-to-understand synthesis and the criteria by which these articles were selected. Are they an exhaustive list? Do they represent a representative sample? What time period do these references cover?
I believe that the article cannot be published without this methodology section, as it does not guarantee that the conclusions drawn by the authors are objective and not just a subjective view.
I believe this problem can be solved and the article can be resubmitted, but this is the editor's decision.
Another aspect I need additional details concerns the model in Figure 1. Do the authors propose this model? This aspect should be explained more clearly: how the authors arrived at this model based on the existing models mentioned above, and what exactly makes this model new and important. And how does the model help the research and the structure of the article?
I hope that these aspects I pointed out are useful for the authors and that they will be solved because the article has good potential.
Good luck!
Author Response

(The authors gave the same response as above.)

Round 2
Reviewer 3 Report
Comments and Suggestions for Authors
The authors have added some information that could contribute to a better understanding of the paper, but have not addressed the recommendations.
For example, the authors have not included a section on methodology in which they explain the research design and approach. Although there is a passage in the introduction, this is neither sufficient nor in the right place. Further explanation has been included in other sections, but there is no coherence and it becomes difficult to discern the methodology.
Even though the authors state that they did not intend an exhaustive review, this fact does not preclude the existence of a selection and review methodology.
The authors also did not explain the role of Figure 1 in organizing and understanding the work.
Limitations and practical contributions are missing.
The authors have responded to the reviews with many explanations "justifying" why they did not follow the suggestions. Still, these do not improve the article's value or make it more useful.
Author Response
Authors' Response to Reviewer’s Comments
We thank the reviewer for their continued consideration of our work. We also acknowledge that the Reviewer wanted to see a formal ‘methodology’ for the review. However, in our responses we went to great lengths to explain that this is not an empirical review, but rather a conceptual review, in which we summarize research trends and then provide an extensive discussion of limitations and shortcomings of that extant research while outlining an agenda for future research to address them. This is in a long tradition of conceptual reviews of work applying self-determination theory in multiple contexts, including those authored by one of the authors of the current review (Bartholomew et al., 2009; Biddle et al., 2001; Deci et al., 2001; Deci & Ryan, 2008; Hagger & Chatzisarantis, 2007, 2008; Ryan & Deci, 2000; Ryan et al., 2021; Sansone & Tang, 2021; Teixeira et al., 2012; Vansteenkiste & Ryan, 2013; Verstuyf et al., 2012) including the workplace (Gagné & Deci, 2005; Grenier et al.; Quinn et al., 2012). These reviews adopt the same or similar approaches to the current review, that is, providing an overview of current work and providing commentary, analysis, and recommendations regarding issues or limitations emerging from the overview. None of them have a formal Method section. They are also explicit in their aims, and do not offer a formal method for doing so, but make explicit the conceptual basis of their review. As an illustration, take Teixeira et al.’s (2012) summary: “In this article, we aimed to explore the topics of motivation and self-regulation from the viewpoint of self-determination theory, in the context of weight management and related behaviors. By doing so, we have offered a somewhat different perspective in the ongoing discussion around promoting sustained behavior changes in the broader literature, which represents one of the most difficult challenges facing health care professionals, behavioral scientists, and the individuals who struggle to make lifestyle change and manage their weight. Specifically…” (p. 10). The authors then go on to summarize four numbered aims and the recommendations arising from them. This review also nicely illustrates the conceptual review approach in a specific behavioral domain, also similar to the current review. In another example, Quinn et al. propose a model of ‘human energy’ in organizational contexts based on a conceptual review of research which they use to justify a model that fills the gaps in research on the subject drawing from multiple theories including self-determination theory, and, importantly, propose future research directions. Justifying their review, they state: “This article begins by looking across the interdisciplinary literatures on human energy to articulate two fundamental definitions of energy, distinguish them from related constructs including burnout, flow, and motivation, and discuss the assumptions that are necessary to build a model of human energy at work. Once this groundwork is laid, each of the six literatures relating to human energy is reviewed. From key findings of each literature, we build a model of human or intra-individual energy which captures the dynamics of human energy over time in a work context. We close the paper by articulating some important directions for future research” (p. 4). Again this illustrates a similar review strategy to the current review, clearly articulating the purpose, approach, and outcomes. We could provide many more examples, and urge the Reviewer to consult the references we have provide for more insight into the approach we have adopted.
However, given the Reviewer’s comments, we thought it important to make sure that our approach is made abundantly clear. As a consequence, we have explicitly cited some of the reviews we noted above in a second revised version of the manuscript. Specifically, we have mentioned how “Our approach follows similar conceptual reviews of research applying self-determination theory in specific contexts, comprising an overview of current work followed by commentary, analysis, and recommendations regarding issues or limitations emerging from the overview” (please see page 3 of the revised manuscript, in green font to differentiate this from the previous responses).
The Reviewer also raises a good point about the need to be explicit about the role of Figure 1. To this end, we state “for clarity, these proposed relations are illustrated in Figure 1, such that the associations between perceived need support and workplace behaviors and outcomes are mediated by need satisfaction and forms of motivation from self-determination theory” (please see page 5 of the revised manuscript, again, in green-colored font).
We would like to politely disagree with the Reviewer on the subject of not discussing the limitations and practical considerations of our review. Throughout the manuscript we have outlined the practical considerations of our review, such as the role that our suggestions may have in improving research studies and developing more effective workplace interventions based on self-determination theory. As a case in point, we state clearly that research aimed at intentions that seek to address the lack of studies examining intervention mechanisms of action will provide better evidence on which organizational leaders can base their interventions, particularly how and why those interventions may or may not work in affecting change in workplace outcomes – see our comment on this, which now appears on page 11-12 of the revised manuscript (we have highlighted it in green so it can easily be found).
Authors’ References
Bartholomew, K. J., Ntoumanis, N., & Thogersen-Ntoumani, C. (2009). A review of controlling motivational strategies from a Self-Determination Theory perspective: Implications for sports coaches. International Review of Sport and Exercise Psychology, 2, 215-233. https://doi.org/10.1080/17509840903235330
Biddle, S. J. H., Chatzisarantis, N. L. D., & Hagger, M. S. (2001). Self-determination theory in sport and exercise. In F. Cury, P. Sarrazin & J. P. Famose (Eds.), Theories de la Motivation et Sport: Etats de la Recherche [Advances in motivation theories and sport] (pp. 19-55). Presses Universitaires de France.
Deci, E. L., Koestner, R., & Ryan, R. M. (2001). Extrinsic rewards and intrinsic motivation in education: Reconsidered once again. Review of Educational Research, 71, 1-27. https://doi.org/10.3102/00346543071001001
Deci, E. L., & Ryan, R. M. (2008). Facilitating optimal motivation and psychological well-being across life's domains. Canadian Psychology, 49(1), 14-23. https://doi.org/10.1037/0708-5591.49.1.14
Gagné, M., & Deci, E. L. (2005). Self-determination theory and work motivation. Journal of Organizational Behavior, 26, 331-362. https://doi.org/10.1002/job.322
Grenier, S., Gagné, M., & O'Neill, T. Self-determination theory and its implications for team motivation. Applied Psychology. https://doi.org/https://doi.org/10.1111/apps.12526
Hagger, M. S., & Chatzisarantis, N. L. D. (2007). Advances in self-determination theory research in sport and exercise. Psychology of Sport and Exercise, 8(5), 597-599. https://doi.org/10.1016/j.psychsport.2007.06.003
Hagger, M. S., & Chatzisarantis, N. L. D. (2008). Self-determination theory and the psychology of exercise. International Review of Sport and Exercise Psychology, 1, 79-103. https://doi.org/10.1080/17509840701827437
Quinn, R. W., Spreitzer, G. M., & Lam, C. F. (2012). Building a sustainable model of human energy in organizations: exploring the critical role of resources. Academy of Management Annals, 6, 337-396. https://doi.org/10.1080/19416520.2012.676762
Ryan, R. M., & Deci, E. L. (2000). Self-determination theory and the facilitation of intrinsic motivation, social development, and well-being. American Psychologist, 55, 68-78. https://doi.org/10.1037//0003-066x.55.1.68
Ryan, R. M., Deci, E. L., Vansteenkiste, M., & Soenens, B. (2021). Building a science of motivated persons: Self-determination theory’s empirical approach to human experience and the regulation of behavior. Motivation Science, 7(2), 97-110. https://doi.org/10.1037/mot0000194
Sansone, C., & Tang, Y. (2021). Intrinsic and extrinsic motivation and self-determination theory. Motivation Science, 7(2), 113-114. https://doi.org/10.1037/mot0000234
Teixeira, P., Silva, M., Mata, J., Palmeira, A., & Markland, D. (2012). Motivation, self-determination, and long-term weight control. International Journal of Behavioral Nutrition and Physical Activity, 9(1), 22. https://doi.org/10.1186/1479-5868-9-22
Vansteenkiste, M., & Ryan, R. M. (2013). On psychological growth and vulnerability: Basic psychological need satisfaction and need frustration as a unifying principle. Journal of Psychotherapy Integration, 23(3), 263-280. https://doi.org/10.1037/a0032359
Verstuyf, J., Patrick, H., Vansteenkiste, M., & Texeira, P. (2012). Motivational dynamics of eating regulation: A Self-Determination Theory perspective. International Journal of Behavioral Nutrition and Physical Activity, 9(1), 21. https://doi.org/10.1186/1479-5868-9-21